# Decreased Associated Risk of Gout in Diabetes Patients with Uric Acid Urolithiasis

**DOI:** 10.3390/jcm8101536

**Published:** 2019-09-25

**Authors:** Chan Jung Liu, Jin Shang Wu, Ho Shiang Huang

**Affiliations:** 1Department of Urology, National Cheng Kung University Hospital, College of Medicine, National Cheng Kung University, Tainan 70403, Taiwan; dragon2043@hotmail.com; 2Department of Family Medicine, National Cheng Kung University Hospital, College of Medicine, National Cheng Kung University, Tainan 70403, Taiwan; jins@mail.ncku.edu.tw; 3Department of Health Management Center, National Cheng Kung University Hospital, College of Medicine, National Cheng Kung University, Tainan 70403, Taiwan; 4Department of Family Medicine, College of Medicine, National Cheng Kung University, Tainan 70101, Taiwan; 5Department of Urology, College of Medicine, National Cheng Kung University, Tainan 70101, Taiwan

**Keywords:** uric acid stone, urolithiasis, gout, diabetes, kidney stone

## Abstract

Uric acid urolithiasis constitutes approximately 7–10% of all urinary stones. Previous studies have revealed that patients with gout do not equally have uric acid stones. Instead, the risk of gout in patients with uric acid stones remains controversial. This study aimed to investigate whether there is different associated risk of gout for diabetes mellitus (DM) and non-diabetes patients with uric acid urolithiasis. Therefore, we examined all baseline chemistries to determine any risk factors or protective factors related to developing gout in patients with uric acid stones. Ninety-nine patients from a single medical center, who had a uric acid component in their stone specimen were enrolled and their medical records were reviewed retrospectively between January 2010 and December 2016. All patients were divided into gout and non-gout groups. Gout was confirmed in 24 patients in this study (24.2%). The proportion of DM was significantly higher in the non-gout group (34.7%) than in the gout group (4.3%, *p* < 0.05). Renal function was decreased and serum triglyceride was higher in patients with gout. Uric acid urolithiasis patients with DM had a lower risk for gout (adjusted odds ratio: 0.08; 95% confidence interval (CI) = 0.01–0.61, *p* = 0.015). In 89 patients with predominant uric acid stones (>50% uric acid composition), the risk for gout was still lower in patients with diabetes than non-diabetes (adjusted odds ratio: 0.08; 95% confidence interval (CI) = 0.01–0.61, *p* = 0.015). These findings suggest that decreased risk of gout is found in uric acid urolithiasis patients with diabetes. Our results imply that patients with uric acid stones should have complete diabetes evaluation before the administration of uric acid controlling medication.

## 1. Introduction

Uric acid is a waste product of purine metabolism; it is also a common component of urinary stones. Uric acid urolithiasis accounts for 7–10% of calculi [1,2]. Accompanying the westernization of diets, especially in developed countries, the prevalence of urolithiasis has increased over recent years. Constantly low urinary pH, hypovolemia and low urinary volume, and hyperuricosuria are pivotal in the pathogenesis of uric acid urolithiasis [1]. The most common cause of hyperuricosuria is dietary imbalance, especially high amounts of purine meats, sugar (fructose), and beer [2]. All of these foods also increase the risk of gout. Gout is a systemic metabolic disorder characterized by monosodium urate crystal accumulation in joints. The association between gout and urolithiasis has been studied; a confirmed history of gout was shown to double the relative risk of incident kidney stones [3]. Although the association between gout and uric acid urolithiasis appears to be indisputable, the incidence of uric acid stones among patients with gout ranges from 10% to 20% [4]. Nearly half of patients with gout have nonuric acid stones [5]. More evidence suggests that excessively acidic urine, which causes titration of urate to the highly insoluble uric acid, may be the key factor causing uric acid urolithiasis. This feature of urine is believed to distinguish gout patients with or without urolithiasis. However, the correlation is still under investigation. Besides, hyperuricemia (defined as serum uric acid ≥ 6 mg/dL (360 μmol/L) in women and ≥ 7 mg/dL (420 μmol/L) in men [6] is strongly associated with metabolic syndrome; the serum uric acid level was found to be positively correlated with the number of metabolic abnormalities [7]. The relationship between uric acid urolithiasis and metabolic syndrome is still controversial. In the current study, we sought to characterize the serum biochemical features and concomitant metabolic syndrome in uric acid stone formers with and without gout, and whether there is a difference in the associated risk of gout between diabetes mellitus (DM) and non-diabetes patients with uric acid urolithiasis. We also examined all baseline chemistries to determine any risk factors or protective factors related to developing gout in uric acid stone formers.

## 2. Materials and Methods

### 2.1. Study Design and Population

We retrospectively reviewed patients from a single tertiary medical center between January 2010 and December 2016. All patients were from urology clinics because of diagnosed urolithiasis. We retrospectively searched for the medical records of all these patients also diagnosed with gout (ICD-9 274 or ICD-10 M10). Gout was diagnosed according to the criteria established by the American College of Rheumatology [8,9]. Patients with uric acid urolithiasis confirmed by stone analysis were included. Every patient was required to have a documented stone containing any amount of uric acid based on the stone analysis. The percentage of uric acid and secondary components for each stone were recorded. All stone analyses were conducted in a single central laboratory. A Perkin–Elmer R × 1-FTIR Fourier transform infrared spectrophotometer was used to analyze the stone composition. Previous studies showed patients with different percentage uric acid compositions had significantly different metabolic profiles [10]. In patients with mixed-stone composition, urinary stones that consisted mostly of uric acid were more representative of uric acid stones. Therefore, patients in our study were then grouped according to percentage uric acid composition: ≤50% and >50% uric acid. We defined those with >50% uric acid as predominant uric acid urolithiasis. None of the patients suffered from secondary gout and other purine metabolic disorders associated with pathological concentrations of serum uric acid (such as a reduced activity of hypoxanthine-guanine phosphoribosyl-transferase and hyperactivity of phosphoribosyl pyrophosphate synthetase 1 resulting in increased levels of xanthine and hypoxanthine) [11].

### 2.2. Data Collection

The reviewed data included patient demographics, body weight, and height on the first visit to clinics, and systemic diseases on medical records (e.g., diabetes mellitus (DM), hypertension (HTN), and cardiovascular disease). Based on their medical history, all the diabetes subjects were type 2 diabetes and none of them were type 1 diabetes. Serum creatinine (Cr), uric acid, triglyceride (TG), cholesterol, and HbA1c were collected before stone intervention. The preoperative urine analysis was also collected, and urine pH values were documented. The estimated glomerular filtration rate (eGFR) was calculated using the MDRD formula: eGFR = 186 × (Serum Cr) ^−1.154^ × (Age) ^−0.203^ (×0.742 if female)(1)

Because waist circumference data was not available in this study, we used the International Diabetes Federation consensus worldwide definition for metabolic syndrome as below: central obesity (if BMI is > 30 kg/m^2^, central obesity can be assumed, and waist circumference does not need to be measured) and any two of the following: triglycerides > 150 mg/dL (1.7 mmol/L), HDL cholesterol < 40 mg/dL (1.03 mmol/L) in males and <50 mg/dL (1.29 mmol/L) in females, systolic/diastolic blood pressure > 130/85 mm Hg, or fasting plasma glucose (FPG) >100 mg/dL (5.6 mmol/L) [12]. This study was approved by the National Cheng Kung University Institutional Review Board (IRB-ER-107).

### 2.3. Statistical Analysis

The results are presented as median and interquartile range unless otherwise noted. Groups were compared using a Mann-Whitney U test for numerical variables and the Fisher exact test for categorical variables. Clinical covariates, which were reported for their cross-sectional associations with gout, included age, gender, body mass index (BMI), creatinine, estimated glomerular filtration rate (eGFR), uric acid, cholesterol, TG, HbA1c, DM, HTN, cardiovascular disease, and hyperlipidemia. Because creatinine and eGFR have multicollinearity effects on multivariable regression analysis, we divided each of them into model 1 and model 2.

All analyses were conducted using SPSS statistical software (versions 16; SPSS Inc., Chicago, CA, USA). Two-tailed *p* < 0.05 was considered statistically significant.

## 3. Results

### 3.1. Demographics and Clinical Characteristics of the Study Population

A total of 99 patients were mainly male (89.9%) with a mean age of 63.40 ± 13.64 and a mean body mass index (BMI) of 25.56 ± 4.25 kg/m^2^. There was no statistically significant difference between males and females with regards to age. Most of the patients were above 60 years old (60.6%). The average uric acid level was 7.15 ± 1.58 mg/dL (425.50 ± 94.13 µmol/L). Gout was confirmed in 24 subjects (24.2%), and all of them were male (Table 1). The demographic characteristics were generally similar in each group. Of the 24 patients with gout, the mean age of these patients was 62.41 ± 16.00 years old, and the mean BMI was 26.71 ± 3.21 kg/m^2^. Renal function appeared to be worse in patients with gout, but neither serum creatinine nor eGFR was significantly different. Significantly more patients without gout had DM (*p* = 0.003). The logistic regression analysis (Table 2) indicated that Cr > 1.5 and eGFR < 45 are risk factors for developing gout. Interestingly, DM seems to be a protective factor against gout development (Adjusted odds ratio (OR) = 0.05, *p* = 0.008). Based on medical records, we further adjusted with two major confounders, which were metabolic syndrome and medications, including thiazide and benzbromazone, that may influence serum uric acid (Appendix A, Appendix A). The lower associated risk of DM for gout was still significant.

### 3.2. Predominant Uric Acid Urolithiasis Patients (Uric Acid > 50%)

Eighty-nine patients had uric acid stones composed of more than 50% uric acid. Of these patients, 21 patients had gout. Table 3 shows the comparison of various clinical parameters between the gout and non-gout uric acid urolithiasis patients. The mean age was similar for the two groups (64.29 ± 13.41 for non-gout and 61.80 ± 15.58 for gout, *p* = 0.483). There were no significant differences among most of the clinical parameters. Importantly, serum creatinine was significantly higher in patients with gout; however, serum levels of creatinine and TG were significantly higher in patients with gout than without gout. Most notably, as compared to patients with gout, those without gout had a higher rate of DM. Using logistic regression analysis (Table 4), we found that Cr > 1.5 and eGFR < 45 are still risk factors to develop gout. Further, uric acid urolithiasis patients with DM were less likely to advance to gout. The correlation still existed even after adjustment.

## 4. Discussion

In the present study, we analyzed the biochemical profiles of uric acid urolithiasis patients with and without gout. Overall, only 24.2% of uric acid urolithiasis patients had gout. We found that uric acid urolithiasis patients without gout were more likely to have DM. The renal functions of these patients were better than those in the other group. DM seems to play a protective role in the development of gout.

Calcium oxalate is the main component in the majority of kidney stones. Generally, uric acid accounts for 7% to 10% of all stone components [13]. Due to the increasing prevalence of obesity and metabolic syndromes, uric acid urolithiasis is affecting increasing numbers of patients. Uric acid stones are known to form from decreased urine output, intestinal alkali loss, and purine overload or overproduction [1]. Of these three reasons, low urinary pH seems to be the most critical factor [14]. A supernormal level of urine uric acid can even be balanced within normal urinary pH values [15]. The solubility of uric acid is controlled by two dissociation constants (pKa). The first pKa of pH 5.5 is the key level in clinical practice. In the current study, the mean urine pH in all patients was 5.6, which was compatible with their physiological characteristics [16].

The precise mechanism of persistent acidic urine in uric acid stone formers is still unclear. Two major factors have been proven to cause acidic urine: reduced renal ammonium (NH_4_+) excretion and increased net acid excretion (NAE) [17,18]. Decreased ammonium secretion results in the loss of urinary buffer. An increase in the concentration of H then significantly decreases urinary pH. Disorders in the enzymes glutaminase and/or glutamate dehydrogenase, which metabolize glutamine into ammonia and ketoglutarate, cause impaired ammonium secretion. Diminished ammonium excretion and increased NAE are also found in diabetic non-stone formers as well as in uric acid stone formers under a fixed metabolic diet [19,20]. A recent study has also revealed that idiopathic uric acid nephrolithiasis can be treated by pioglitazone, which is a peroxisome proliferator-activated receptor gamma (PPAR-γ) agonist to treat diabetes [21]. Thus, it is very clear that there is a close connection between uric acid stone and DM. In addition to DM, obesity is also associated with acidic urine, and is a strong risk factor of urolithiasis [22]. It is worth mentioning that mean BMI in previous uric acid urolithiasis studies was around 30 kg/m^2^ [23,24]. However, mean BMI in the current study was only 25.6 ± 6.7 kg/m^2^. Interestingly, Takeuchi et al. have shown that liver computed tomography (CT) Hounsfield unit value was negatively associated with BMI in urolithiasis patients and they considered that other mechanisms unassociated with fatty liver may be involved in urolithiasis in non-obese patients [25]. Further studies are warranted to determine the role of fatty liver in uric acid urolithiasis formation.

There are many important findings that resulted from the current study. First, uric acid urolithiasis patients with DM were less likely to have gout. Hyperuricosuria may play a key role. Patients with DM are known to have hyperuricosuria as well as acidic urine [19,26]. Previous research has revealed that glycosuria induced by sodium-glucose cotransporter 2 (SGLT2) inhibitors can subsequently lower uric acid level [27]. There was a significantly negative association between uric acid level and urine glucose. Our previous study also confirms this finding [28]. Uric acid level was inversely associated with fasting plasma glucose in diabetic patients. All this evidence suggests that patients with DM are prone to excrete uric acid into urine, which then provides a favorable environment to develop uric acid stones. Moreover, gouty arthritis forms when patients have hyperuricemia and subsequently develop monosodium urate crystal deposition in joints [29]. However, the activation of gouty arthritis cannot be stimulated by crystals alone. It involves co-stimulation with free fatty acids or lipopolysaccharide to release IL1β [30]. This process is initiated by a change in the concentration of free fatty acid or serum uric acid. It is believed that a constant serum concentration of uric acid is important to prevent formation of gouty arthritis. Thus, patients with DM may have relatively stable serum uric acid through uninterrupted urinary uric acid excretion. The possibility of experiencing gouty arthritis may consequently be lower. This could also explain why the uric acid urolithiasis patients with gout had worse renal functions. Since uric acid is excreted mostly by the kidney, a rise in serum uric acid is known to occur as the glomerular filtration rate falls. Hence, a decrease in the glomerular filtration rate (GFR) is inevitably accompanied by more frequent gouty arthritis. Recently, Bobulescu et al. compared uric acid urolithiasis patients with BMI-matched controls [18]. Serum BUN and creatinine were all significantly higher in the uric acid urolithiasis group. All these findings suggest that urinary uric acid excretion may not only induce urolithiasis formation but may also decrease renal function. Finally, serum uric acid accumulation or instability contribute to gout formation. However, further investigation is necessary to determine the exact pathophysiology.

Previous studies have revealed that hyperuricosuria patients might have stone components other than uric acid [5]. Nearly 10% of calcium stone formers have hyperuricosuria as an isolated metabolic abnormality [31]. Although the mechanism by which hyperuricosuria prompts CaOx has not been fully revealed, overly acidic urine provides a favorable environment to develop CaOx stones. This entity has been termed “hyperuricosuric CaOx urolithiasis”, which is different from uric acid stones [32]. Hyperuricosuric CaOx has been described according to different definitions [23,26]. In contrast, uric acid stone formation without secondary causes (such as chronic diarrhea, dehydration, or purine food overload) has been defined as idiopathic uric acid urolithiasis or gouty diathesis [33]. The underlying disorder in hyperuricosuric CaOx is believed to originate from consumption of high-purine foods. The underlying mechanism in idiopathic uric acid urolithiasis is believed to be gout even though some recent evidence does not support this assumption [18,23]. Because many stones have multiple components, these two types of stones may all be identified with uric acid components through a stone analysis. On the other hand, many possible secondary causes may promote uric acid stones or hyperuricosuric CaOx stone formation, such as gout and DM; every single cause may independently contribute to stone growth. Thus, it is possible that one may serve as an inhibitory factor on another. Accordingly, our results should be interpreted with caution. Finally, there is an interesting but unanswered question derived from our study. Among all lipid profiles, only uric acid urolithiasis with gout patients had significantly higher TG. Previous studies also revealed that idiopathic uric acid stone patients have significantly higher TG compared to controls [17,18,22,34,35]. Therefore, this evidence supports our hypothesis that the patients in the current study who had gout were likely to be idiopathic uric acid stone formers. Besides, previous studies found that patients with significantly higher TG had higher urinary uric acid than patients with low TG [35,36,37]. The pathogenic relation between TG and uric acid urolithiasis remains to be determined.

There are some limitations and unanswered findings. First, this study is retrospective in nature and is thus subject to potential selection biases. Second, we lacked the 24-h urine biochemistry necessary for a full stone risk analysis. Third, our study recruited a relatively small number of cases. Fourth, urolithiasis can be related to both hyperuricemia and hypouricemia, though the prevalence of hypouricemia is low. Among hypouricemia, elevated excretion fraction of uric acid (EF-UA) can be used to differentiate renal hypouricemia (RHUC) from hereditary xanthinuria (HX) [38]. RHUC is strongly associated with urolithiasis and kidney injury [39]; its clinical presentation is similar to our cases in some respects. However, based on our review of medical records, we could not identify these subjects from our database. In the future, we intend to collect more cases to prove our observations.

## 5. Conclusions

Only 24% of the uric acid urolithiasis patients had gout concomitantly. Over one third of uric acid urolithiasis patients with DM did not develop gout. There was less associated risk of gout in uric acid urolithiasis patients with diabetes.

## Figures and Tables

**Table 1 jcm-08-01536-t001:** Difference in clinical features and laboratory findings between uric acid urolithiasis patients with and without gout.

	Without Gout	With Gout	*p* Value
	(*n* = 75)	(*n* = 24)	
	*n* (%)	*n* (%)	
**Age**			
≤60	27 (36.00)	12 (50.00)	0.326
>60	48 (64.00)	12 (50.00)	
median (IQR)	65.00 (57.00, 72.00)	60.50 (49.50, 73.00)	0.363 ^a^
	(*n* = 75)	(*n* = 24)	
**BMI (kg/m^2^)**			
≤25	35 (58.33)	6 (27.27)	0.025 ^b^
>25	25 (41.67)	16 (72.73)	
mean ± SD	25.14 ± 4.52	26.70 ± 3.21	0.141
	(*n* = 60)	(*n* = 22)	
**Uric acid (mg/dL)/(µmol/L)**			
mean ± SD	7.14 ± 1.55/424.50 ± 92.23	7.19 ± 1.71/427.88 ± 101.56	0.905
	(*n* = 38)	(*n* = 16)	
**Urine pH**			
median (IQR)	6.00 (5.00, 6.00)	6.00 (5.25, 6.00)	0.541 ^a^
	(*n* = 72)	(*n* = 24)	
**Creatinine (mg/dL)**			
≤1.5	56 (75.68)	13 (54.17)	0.08
>1.5	18 (24.32)	1 (45.83)	
median (IQR)	1.18 (0.96, 1.46)	1.38 (1.12, 1.84)	0.052 ^a^
	(*n* = 74)	(*n* = 24)	
**eGFR (mL/min/1.73 m^2^)**			
<45	14 (18.92)	10 (41.67)	0.048
≥45	60 (81.08)	14 (58.33)	
median (IQR)	65.67 (50.31, 77.07)	55.56 (39.03, 70.37)	0.167 ^a^
	(*n* = 74)	(*n* = 24)	
**Cholesterol (mg/dL)**			
mean ± SD	176.27 ± 51.95	185.83 ± 37.38	0.566 ^b^
	(*n* = 30)	(*n* = 12)	
**Triglyceride (mg/dL)**			
median (IQR)	101.00 (75.00, 122.00)	157.00 (124.50, 176.50)	0.021 ^a^
	(*n* = 30)	(*n* = 12)	
**HbA1C (%)**			
median (IQR)	6.00 (5.70, 6.90)	5.80 (5.40, 6.60)	0.297 ^a^
	(*n* = 34)	(*n* = 15)	
**Diabetes mellitus**			
No	49 (65.33)	23 (95.83)	0.003 ^b^
Yes	26 (34.67)	1 (4.17)	
**Hypertension**			
No	41 (54.67)	11 (45.83)	0.603
Yes	34 (45.33)	13 (54.17)	
**Cardiovascular disease**			
No	61 (81.33)	22 (91.67)	0.344 ^b^
Yes	14 (18.67)	2 (8.33)	
**Hyperlipidemia**			
No	18 (60.00)	5 (41.67)	0.462
Yes	12 (40.00)	7 (58.33)	

Data presented as means ± standard deviation or percentage (%); ^a^ Mann–Whitney U test; ^b^ Fisher’s exact test. Abbreviations: BMI—Body mass index; HbA1c—Glycated hemoglobin A1c; eGFR—Estimated glomerular filtration rate.

**Table 2 jcm-08-01536-t002:** Logistic regression analysis of risk factors for gout among uric acid urolithiasis patients.

	Crude OR (95 % CI)	*p*-Value	Adjusted OR (Model 1) (95 % CI)	*p*-Value	Adjusted OR (Model 2) (95 % CI)	*p*-Value
**Age**	0.99 (0.95–1.02)	0.393				
**Age group**						
≤60	Ref.					
>60	0.56 (0.22–1.42)	0.225				
**Gender**						
Female	Ref.					
Male	9.65 (0.49–189.89)	0.136 ^a^				
**BMI (kg/m^2^)**	1.10 (0.97–1.24)	0.142				
**Uric acid (mg/dL)**	1.02 (0.70–1.49)	0.903				
**Urine pH**	1.33 (0.69–2.58)	0.392				
**Creatinine (mg/dL)**	1.48 (0.78–2.78)	0.231				
**Creatinine group**						
≤1.5	Ref.				Ref.	
>1.5	2.63 (1.01–6.89)	0.049			5.09 (1.40–18.56)	0.014
**eGFR (mL/min/1.73 m^2^)**	0.99 (0.97–1.01)	0.233				
**eGFR group**						
≥45	Ref.		Ref.			
<45	3.06 (1.13–8.31)	0.028	8.40 (1.85–38.06)	0.006		
**Cholesterol (mg/dL)**	1.00 (0.99–1.02)	0.556				
**Triglyceride (mg/dL)**	1.01 (1.00–1.02)	0.231				
**HbA1C (%)**	1.07 (0.68–1.67)	0.774				
**Diabetes mellitus**						
No	Ref.		Ref.		Ref.	
Yes	0.08 (0.01–0.64)	0.017	0.04 (0.01–0.42)	0.007	0.05 (0.01–0.47)	0.008
**Hypertension**						
No	Ref.					
Yes	1.43 (0.57–3.59)	0.452				
**Cardiovascular disease**						
No	Ref.					
Yes	0.40 (0.08–1.88)	0.245				
**Hyperlipidemia**						
No	Ref.					
Yes	2.10 (0.54–8.19)	0.285				

Adjusted OR model 1: using eGFR as renal function representative variable. Adjusted OR model 2: using creatinine as renal function representative variable. ^a^ Firth logistic regression. Abbreviations: BMI—Body mass index; HbA1c—Glycated hemoglobin A1c; eGFR—Estimated glomerular filtration rate.

**Table 3 jcm-08-01536-t003:** Differences in clinical features and laboratory findings between predominant uric acid urolithiasis patients (uric acid > 50%) with and without gout.

	Without Gout	With Gout	*p* Value
	(*n* = 68)	(*n* = 21)	
	*n* (%)	*n* (%)	
**Age**			
≤60	24 (35.29)	10 (47.62)	0.448
>60	44 (64.71)	11 (52.38)	
median (IQR)	65.50 (57.00, 73.50)	63.00 (49.00, 71.00)	0.341 ^a^
	(*n* = 68)	(*n* = 21)	
**Gender**			
Female	11 (16.18)	0 (0.00)	0.060 ^b^
Male	57 (83.82)	21 (100.00)	
**BMI (kg/m^2^)**			
≤25	34 (59.65)	6 (31.58)	0.063
>25	23 (40.35)	13 (68.42)	
mean ± SD	25.04 ± 4.58	26.50 ± 3.34	0.205
	(*n* = 57)	(*n* = 19)	
**Uric acid (mg/dL)/(µmol/L)**			
mean ± SD	7.28 ± 1.45/432.88 ± 86.35	7.23 ± 1.76/430.24 ± 104.67	0.926
	(*n* = 36)	(*n* = 15)	
**Urine pH**			
median (IQR)	5.50 (5.00, 6.00)	5.50 (5.00, 6.00)	0.775 ^a^
	(*n* = 67)	(*n* = 21)	
**Creatinine (mg/dL)**			
≤1.5	51 (76.12)	11 (52.38)	0.071
>1.5	16 (23.88)	10 (47.62)	
median (IQR)	1.16 (0.94, 1.46)	1.47 (1.20, 1.84)	0.037 ^a^
	(*n* = 67)	(*n* = 21)	
**eGFR (mL/min/1.73 m^2^)**			
<45	13 (19.40)	9 (42.86)	0.061
≥45	54 (80.60)	12 (57.14)	
median (IQR)	67.90 (50.56, 77.43)	50.94 (38.98, 69.56)	0.123 ^a^
	(*n* = 67)	(*n* = 21)	
**Cholesterol (mg/dL)**			
≤220	23 (76.67)	10 (83.33)	1.000 ^b^
> 220	7 (23.33)	2 (16.67)	
mean ± SD	176.27 ± 51.95	185.83 ± 37.38	0.566
	(*n* = 30)	(*n* = 12)	
**Triglyceride (mg/dL)**			
≤140	24 (80.00)	5 (41.67)	0.026 ^b^
>140	6 (20.00)	7 (58.33)	
median (IQR)	101.00 (75.00, 122.00)	157.00 (124.50, 176.50)	0.021 ^a^
	(*n* = 30)	(*n* = 12)	
**HbA1C (%)**			
≤7	26 (78.79)	14 (93.33)	0.406 ^b^
>7	7 (21.21)	1 (6.67)	
median (IQR)	6.00 (5.70, 6.90)	5.80 (5.40, 6.60)	0.257 ^a^
	(*n* = 33)	(*n* = 15)	
**Diabetes mellitus**			
No	42 (61.76)	20 (95.24)	0.003 ^b^
Yes	26 (38.24)	1 (4.76)	
**Hypertension**			
No	34 (50.00)	9 (42.86)	0.747
Yes	34 (50.00)	12 (57.14)	
**Cardiovascular disease**			
No	55 (80.88)	19 (90.48)	0.506 ^b^
Yes	13 (19.12)	2 (9.52)	
**Hyperlipidemia**			
No	18 (60.00)	5 (41.67)	0.462
Yes	12 (40.00)	7 (58.33)	

Data presented as means ± standard deviation or percentage (%); ^a^ Mann-Whitney U test; ^b^ Fisher’s exact test. Abbreviations: BMI—Body mass index; HbA1c—Glycated hemoglobin A1c; eGFR—Estimated glomerular filtration rate.

**Table 4 jcm-08-01536-t004:** Logistic regression analysis of risk factors for gout among predominant uric acid urolithiasis patients (uric acid > 50%).

	Crude OR (95 % CI)	*p*-Value	Adjusted OR (Model 1) (95 % CI)	*p*-Value	Adjusted OR (Model 2) (95 % CI)	*p*-Value
**Age**	0.98 (0.95–1.02)	0.304				
**Age group**						
≤60	Ref.					
>60	0.60 (0.22–1.62)	0.312				
**Gender**						
Female	Ref.					
Male	8.60 (0.43–172.50)	0.160 ^a^				
**BMI (kg/m^2^)**	1.08 (0.96–1.23)	0.205				
**BMI group**						
≤25	Ref.		Ref.		Ref.	
>25	3.20 (1.06–9.65)	0.039	3.42 (0.96–12.22)	0.059	3.60 (1.01–12.77)	0.048
**Uric acid (mg/dL)**	0.98 (0.66–1.46)	0.924				
**Urine pH**	1.20 (0.62–2.31)	0.591				
**Creatinine (mg/dL)**	1.51 (0.79–2.88)	0.213				
**Creatinine group**						
≤1.5	Ref.				Ref.	
>1.5	2.90 (1.04–8.07)	0.042			5.56 (1.43–21.62)	0.013
**eGFR (mL/min/1.73 m^2^)**	0.99 (0.97–1.01)	0.201				
**eGFR group**						
≥45	Ref.		Ref.			
<45	3.12 (1.08–8.95)	0.035	7.83 (1.69–36.33)	0.009		
**Cholesterol (mg/dL)**	1.00 (0.99–1.02)	0.556				
**Triglyceride (mg/dL)**	1.01 (1.00–1.02)	0.231				
**HbA1C (%)**	1.05 (0.67–1.66)	0.817				
**Diabetes mellitus**						
No	Ref.		Ref.		Ref.	
Yes	0.08 (0.01–0.64)	0.017	0.05 (0.01–0.46)	0.008	0.05 (0.01–0.48)	0.009
**Hypertension**						
No	Ref.					
Yes	1.33 (0.50–3.58)	0.568				
**Cardiovascular disease**						
No	Ref.					
Yes	0.45 (0.09–2.16)	0.315				
**Hyperlipidemia**						
No	Ref.					
Yes	2.10 (0.54–8.19)	0.285				

Adjusted OR model 1: using eGFR as renal function representative variable. Adjusted OR model 2: using creatinine as renal function representative variable. ^a^ Firth logistic regression. Abbreviations: BMI—Body mass index; HbA1c—Glycated hemoglobin A1c; eGFR—Estimated glomerular filtration rate.

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
