# Peer review of "Decreased Associated Risk of Gout in Diabetes Patients with Uric Acid Urolithiasis"

_jcm, 2019, doi:10.3390/jcm8101536_

Round 1
Reviewer 1 Report
The data are convincing and the discussion appropriate. The methods are sound and the results presented are reliable. However, the manuscript would be strengthened considerably by revision in response to the following comments:
Was cohort controlled for several potential confounders, such as the types of medications that may influence serum uric acid concentrations?
Patients suffering from secondary gout and other purine metabolic disorders associated with pathological concentrations of SUA (such as a reduced activity of hypoxanthine-guanine phosphoribosyl-transferase and superactivity of phosphoribosyl pyrophosphate synthetase 1 resulting in increased levels of xanthine and hypoxanthine) were excluded?
Urolithiasis is strong clinical sign not only in hyperuricemia but also in hypouricemia. Hypouricemia is a relatively rare condition, occurring in about 0.15–3.3% of the general population and 1.2–4% in hospitalized patients. Excretion fraction of uric acid (EF-UA) is a key biochemical marker for a diferential diagnosis of primary hypouricemia. Markedly elevated EF-UA suggests renal hypouricemia (RHUC) while lower or normal EF-UA suggests hereditary xanthinuria (Urolithiasis. 2015 Feb;43(1):61-7).
The high incidence of RHUC1 (OMIM #220150) has been reported in the Asia region and is attributed to the high frequency (2.30–2.37 %) of the c.774G>A (p.W258X) and (0.40 %) c.269G>A (p.R90H) in SLC22A12 gene among Japanese and Koreans, which is indicative of a founder mutation on the Asian continent. However, was reported unexpectedly high frequency of SLC22A12 variants causing RHUC1 in the Roma population (the largest and the most widespread ethnic minority of Europe): frequencies of the c.1245_1253del and c.1400C>T dysfunctional variants were 1.87 and 5.56 %, respectively (Nucleosides Nucleotides Nucleic Acids. 2016 Dec;35(10-12):529-535).
More than 20 patients with RHUC, caused by heterozygous defects in the SLC2A9 gene (OMIM #612076, RHUC2), coding for GLUT9, have been described. Homozygous and/or compound heterozygous loss-of-function mutations in SLC2A9, which are responsible for severe hypouricemia (and very often complicated by urolithiasis, nephrolithiasis and AKI), have also been reported.
Several studies explored the links between increased serum UA concentration and various components of metabolic syndrome. However, the serum UA concentration correlates positively (p<0.001) also with the number of MS criteria (PLoS One. 2014 May 14;9(5):e97646).
not all readers will be familiar with the units mg/dL. For this reason, the reference ranges should be given, these will of course vary with age and sex, include conversion in µmol/l.
Reviewer 2 Report
The manuscript reported that there is decreased risk of gout in patients with diabetes and having uric acid urolithiasis. However several other studies have reported that there is a controversial debate around this understanding.The convclusion of the study has supported the study.
Specific comments:
Authors have considered which diabetis patients both type I and Type II? how the authors have handled the metabolic syndrome patients? have the authors considered those patients as well in the study?Author Response
Please see the attachment.

Reviewer 3 Report
Title: Decreased associated risk of gout in diabetes patients with uric acid urolithiasis
Dear the authors,
General comment:
This study investigated the relevance of gout in diabetesand uric acid urolithiasis.
This is a nicely written manuscript that provides useful information for uric acid urolithiasis.
However, there are some concerns as follows:
Major comment:
I think gender is an important consideration for diseases in metabolic syndrome like uric acid and diabetes. In addition, fatty liver may also be a factor. Please refer to the following literature.
Clinical characteristics in urolithiasis formation according to body mass index.
Takeuchi H, Aoyagi T.
Biomed Rep. 2019 Jul;11(1):38-42. doi: 10.3892/br.2019.1220.
It is recommended to add a discussion that includes these items, or add an explanation to these discussions.
Sincerely,
Round 2
Reviewer 1 Report
The authors have made methodological, result and textual amendments to the paper which strengthen it.
Reviewer 2 Report
can be accepted in current format